# Enhancing Knee MR Image Clarity through Image Domain Super-Resolution Reconstruction

**Vishal Patel [1], Alan Wang [1,2,\*], Andrew Paul Monk [1] and Marco Tien-Yueh Schneider [1]**

[1]   Auckland Bioengineering Institute, The University of Auckland, Auckland 1010, New Zealand
[2]   Faculty of Medical and Health Sciences, The University of Auckland, Auckland 1010, New Zealand
\*   Correspondence: alan.wang@auckland.ac.nz; Tel.: +64-99234402

**Abstract:** This study introduces a hybrid analytical super-resolution (SR) pipeline aimed at enhancing the resolution of medical magnetic resonance imaging (MRI) scans. The primary objective is to overcome the limitations of clinical MRI resolution without the need for additional expensive hardware. The proposed pipeline involves three key steps: pre-processing to re-slice and register the image stacks; SR reconstruction to combine information from three orthogonal image stacks to generate a high-resolution image stack; and post-processing using an artefact reduction convolutional neural network (ARCNN) to reduce the block artefacts introduced during SR reconstruction. The workflow was validated on a dataset of six knee MRIs obtained at high resolution using various sequences. Quantitative analysis of the method revealed promising results, showing an average mean error of $1.40 \pm 2.22\%$ in voxel intensities between the SR denoised images and the original high-resolution images. Qualitatively, the method improved out-of-plane resolution while preserving in-plane image quality. The hybrid SR pipeline also displayed robustness across different MRI sequences, demonstrating potential for clinical application in orthopaedics and beyond. Although computationally intensive, this method offers a viable alternative to costly hardware upgrades and holds promise for improving diagnostic accuracy and generating more anatomically accurate models of the human body.

**Keywords:** super-resolution; magnetic resonance images (MRI); image reconstruction; machine learning; convolutional neural networks (CNN)

## 1. Introduction

Non-invasive medical images are the source of information from which we create and test anatomically accurate models of the human body. Magnetic resonance images (MRIs) are the favoured imaging technique to view soft tissues, but its ability to generate 3D anatomical images is constrained by scan time, patient motion and scanning parameters. During the imaging process, the patient must remain still to prevent blurring and motion artefacts appearing in the image, which is more likely to occur during a long scanning time. Improved spatial resolution may be achieved at the expense of reduced signal-to-noise ratio (SNR) and/or increased scan time. As a result, a trade-off is typically made between these factors in clinical MRI acquisition, which results in MRI stacks that have a high in-plane resolution (x, y-direction) and low out-of-plane resolution (z-direction) [1–3]. In the clinic, three orthogonal MRIs are typically collected in order to maximise the resolution and visible detail in each of the axial, coronal and sagittal planes. However, this is an issue as each slice in the image, comprised of small 3D units called voxels, has an associated slice thickness and/or a gap [1–3]. This causes partial volume effects that make it difficult to obtain detailed morphological information, reducing the accuracy and confidence of the models we derive from the images.

In recent years, there have been significant advancements in hardware and imaging technologies that have enabled the acquisition of good high-resolution images. Tech-

niques such as parallel imaging have resolved some of the shortcomings of old hardware by providing faster acquisition time and higher resolution images. Newer high-magnetic-strength 3.0 T MR scanners can obtain higher-SNR images with voxel sizes of 0.3 mm × 0.3 mm × 0.5 mm. However, these technologies are expensive; for example, new MR scanners can cost up to ~USD 3M and are not prevalent [1,4], motivating the need for techniques that can improve detail and resolution in existing hardware without excessive additional costs. Post-acquisition image processing techniques, such as super-resolution (SR), have been used to alleviate some of these issues. SR techniques have the advantage of being able to increase the resolution of images without having to purchase new, improved hardware, and some techniques do not require additional increase to the scanning time.

SR is a class of techniques used to enhance the resolution of images. This can be achieved in the k-space/frequency domain or in the image domain [1]. However, image domain methods are more flexible and tend to perform better than k-space methods as they allow incorporation of complex models of motion, slice profile, slice geometry, and point spread function [1]. Various SR methods have been developed and include inverse, image fusion, machine learning, and interpolation approaches.

Inverse SR techniques involve exploiting the general form of the image acquisition model. This SR approach is typically viewed as an ill-posed inverse problem where the aim is to recover the high-resolution image given a set of low-resolution images. This is achieved by performing optimisation to minimise the least squared cost function. This minimisation often leads to an unstable solution as the problem is under-constrained. Therefore, regularisation is employed to ensure uniqueness in the minimisation process [5]. A number of approaches are used to perform this regularisation such as the iterative back projection method [5–9], the maximum a posteriori method [5,10] and the projection onto convex sets (POCS) method [5,11,12]. These methods generate reasonable outputs, though each technique is accompanied by its own particular set of limitations such as computational cost, requiring prior information about the desired solution or requiring prior knowledge on the noise. This means these techniques can vary in performance depending on the situation [5].

Image fusion SR techniques increase the through-plane resolution by combining information from several MRI stacks. These stacks can be obtained in parallel by shifting the acquisition space in the slice-select direction by a known subpixel distance [5] or, alternatively, in orthogonal planes [13–17]. The images are then combined using a reconstruction algorithm to synthesise a single high-resolution 3D image dataset with isotropic voxel size. These approaches can be very effective in generating high-quality images and have better attenuation of partial volume effects. The downside is that multiple MR scans need to be obtained, which can be time-consuming and clinically challenging. Some reconstruction algorithms average the intensity values which can introduce blurring and decrease contrast. Interpolation is also often used, which reduces the fidelity of the image and can introduce interpolation artefacts.

Machine learning-based SR techniques have developed rapidly in recent years. These methods aim to establish a correlation between high- and low-resolution images [5,18–22]. However, as these techniques synthesise new data based on previous training, they are susceptible to errors from poor or inappropriate training and over-fitting. Additionally, we still do not have complete confidence in whether output accurately reflects the true image.

Interpolative SR techniques involve using interpolation algorithms such as nearest neighbour, bicubic and bilinear algorithms to reconstruct a high-resolution image by estimating the voxel values between the slices [21,23,24]. These methods are computationally efficient at producing HR images, but they can result in images with blurred edges, interpolation artefacts, and overly smooth images that do not capture the fine details.

Studies have demonstrated that deep learning is effective in reducing image artefacts [25–28]. Therefore, neural networks have the potential to be used post SR to correct for any artefacts that are synthesised in the SR image.

Considering the trade-offs present with existing methods, we propose a hybrid analytical SR method that combines information from the three orthogonal planes that are typically obtained clinically. This method has minimal reliance on interpolation, so it reduces blurring and artefacts in the reconstructed image. Additionally, in contrast to methods that involve machine learning during reconstruction [18–20], this method limits deep learning to only the post-reconstruction step to correct for any artefacts. This eliminates prediction in the reconstruction step and, in turn, minimises prediction throughout the overall algorithm. We detail the method below and present quantitative and qualitative results on a set of six knee MRIs.

## 2. Methods

The super-resolution method developed in this study was an image-domain post-processing algorithm that consisted of three main steps (Figure 1). The first step was a pre-processing step to prepare the clinical images for the super-resolution algorithm. This involved re-slicing and registering the image stacks. The second step was the super-resolution step, where the three image stacks were combined to reconstruct a higher-resolution image stack. The final step was a post-processing step that used a neural network to reduce block artefacts introduced in the super-resolution step. We describe each of the three steps in the following three sections and detail the in the fourth section the validation study performed to evaluate the method.

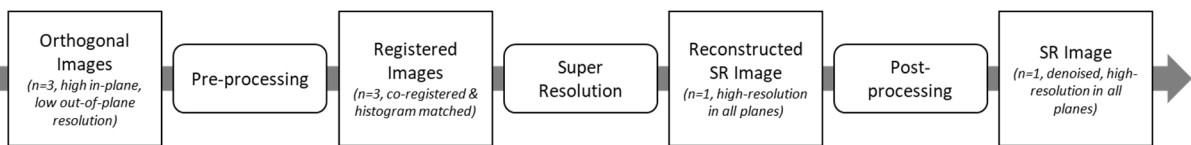

**Figure 1.** Overview of the super-resolution workflow. The workflow consists of 3 main steps: pre-processing, super-resolution, and post-processing.

### 2.1. Preprocessing

The image stacks of the knee were first prepared for the super-resolution algorithm. Images were assumed to come from clinical magnetic resonance (MR) scanners that typically have an in-plane resolution of around 0.5 mm and a low out-of-plane resolution of >3.0 mm and consisted of images obtained in three orthogonal planes (axial, coronal, and sagittal). Although an SR image can be reconstructed by only using two orthogonal planes, we found that the additional image data provided by a third plane resulted an output image that is more consistent and has greater quality when viewed in all directions. A comparison can be found in Appendix A.1.

Prior to registration, histogram matching was performed using the SciPy Python image library to normalise the intensity values of the axial and coronal image stacks to the target (sagittal) image stack.

One of the image stacks was selected as the target for registration. The sagittal plane is the orientation which provides the best view of knee structures such as the ACL, and so is typically the plane used to analyse and segment these structures. Therefore, the sagittal image stack was selected as the target to better preserve the detail of the knee structures. This target was resliced to be isotropic such that the out-of-plane plane resolution matched the in-plane resolution. This was performed to provide a high-resolution target for registration and to determine the image stack dimensions for the super-resolution algorithm.

The axial and coronal plane stacks were individually registered to the sagittal images in the 3D slicer using the general registration BRAINS module [29,30]. The cost metric was changed to normalised correlation as all our images were of the same modality and MR sequence. This registration method aimed to maximise the image intensity data across the imaging planes and in turn help to reduce motion-related errors that may have been introduced during imaging. Rigid and affine registration were selected as the registration phases and were performed in that order, such that if the operation failed to converge

with rigid registration, affine registration would be employed to involve higher degrees of freedom (DOF). All other parameters were left as the default.

During registration, the axial and coronal images were resampled via linear interpolation to the same resolution and dimensions as the target sagittal image. Therefore, we resliced the registered axial and coronal images to their original resolutions to reduce the search space and minimise the computational cost of reconstruction in the superresolution algorithm.

A detailed list of the registration parameters can be accessed in Appendix A.2.

### 2.2. Super-Resolution

The super-resolution algorithm developed calculates each voxel intensity in the reconstructed image based on a linear inverse weighted average of the sagittal, axial, and coronal images.

The inputs to this algorithm include the registered and normalised image stacks (axial, coronal, and sagittal stacks) and a template image containing the corresponding centroid coordinates of each voxel that are reconstructed during SR. This template is initialised with dimensions and voxel centroids equal to the target isotropic sagittal stack. For each voxel in the template, the algorithm finds the closest voxel in each of the three stacks and calculates the intensity by averaging the three intensity values based on the vector magnitude to the template voxel being reconstructed. Therefore, the closer a voxel is to the template, the higher the weighting and hence contribution to the intensity value (Equations (1)–(4)). A worked example to calculate template intensity values can be found in Appendix A.3.

$$v_{template_{ijk}} = \sum (w_{plane} \times v_{plane}), \tag{1}$$

$$w_{plane} = \frac{d_{Total_{ijk}} - d_{plane}}{2d_{Total_{ijk}}}, \tag{2}$$

$$d_{plane} = d(c_{template}, c_{plane_{closest}}) = \|c_{template} - c_{plane_{closest}}\| =$$
$$\sqrt{(c_{template_x} - c_{plane_x})^2 + (c_{template_y} - c_{plane_y})^2 + (c_{template_z} - c_{plane_z})^2}, \tag{3}$$

$$d_{Total_{ijk}} = \sum d_{plane}, \tag{4}$$

where $v$ is voxel intensity. Voxel intensity for each voxel in the template is calculated as the weighted sum of voxel intensity in each plane (Equation (1)). The weights for each voxel of each plane are obtained using Equation (2), where $d$ is the distance between the centroids of the closest voxel in the specific image plane (Equation (3)) and the template voxel, $d_{Total_{ijk}}$ is the total distance (Equation (4)). Subscript $ijk$ refers to the position of the voxel in the MRI 3D image array, where $i$ denotes the row in the MR image slice, $j$ denotes the column in the MR image slice and $k$ denotes the number of the MR image slice itself.

### 2.3. Post-Processing—Artefact Reduction Convolutional Neural Network

To reduce block artefacts that were introduced in the super-resolution step, we developed an artefact reduction convolutional neural network (ARCNN) based on the work of Yu et al. [25].

The ARCNN was developed to de-noise 2D MRI slices rather than the entire 3D volume at once. To train the ARCNN, we used the noisy sagittal slices from the reconstructed SR image and their corresponding slices in the original sagittal image as the input and target images, respectively.

We performed a couple of pre-processing steps before training the ARCNN. First, all the training images were down-sampled to a common size of 256 by 256 for the input and target images. This had the added benefit of reducing the computational cost without sacrificing too much detail in the image. Second, we normalised the intensity values of the images between 0 and 1. This helped to speed up learning and led to a faster convergence.

The convolutional neural network was developed with a 5-layer architecture, similar to the Fast-ARCNN detailed by Yu et al. [25], using the tensorflow keras Python library, which included layers for (i) feature extraction, (ii) shrinking, (iii) feature enhancement, (iv) mapping, and (v) reconstruction (Table 1).

**Table 1.** Summary of ARCNN architecture parameters.

| Layer | Type | Filters (i.e., Number of Output Channels) | Kernel (i.e., Size of the Convolution Filter) | Stride | Padding | Activation |
|---|---|---|---|---|---|---|
| (i) Feature extraction | Conv2D | 64 | $9 \times 9$ | 1 | Zero padding | ReLU |
| (ii) Shrinking | Conv2D | 32 | $1 \times 1$ | 1 | No padding | ReLU |
| (iii) Feature enhancement | Conv2D | 32 | $7 \times 7$ | 1 | Zero padding | ReLU |
| (iv) Mapping | Conv2D | 64 | $1 \times 1$ | 1 | No padding | ReLU |
| (v) Reconstruction | Conv2D Transpose | 1 | $7 \times 7$ | 1 | Zero padding | None |

The ARCNN developed by Yu et al. was demonstrated to be very effective in denoising image artefacts so we did not greatly deviate from their design. However, we made some slight modifications that we believed would help the network to perform better for our application. The key distinctions were that we maintained a stride length of 1 for each layer compared to the stride length of 2 used in their network in order to better preserve spatial information. We also employed the ReLU activation function rather than PreLU due to its greater simplicity and computational efficiency. Finally, our model used an Adam optimiser rather than stochastic gradient descent due to the optimiser's ability to adaptively adjust its learning rate and thus achieve faster convergence.

The feature extraction layer (i) uses 9 by 9 filters to extract overlapping patches from the input image and represents each patch as a feature map. At this layer, the input image was separated into 64 feature maps. To accelerate learning, the shrinking layer (ii) mapped the high-dimensional features to a lower-dimensional feature space (i.e., 64 feature maps to 32). However, because of the block artefacts, the extracted features appeared noisy and so the following feature enhancement layer (iii) used 7 by 7 filters to further denoise the feature maps. The mapping layer (iv) then mapped the cleaned feature maps to a higher-dimensional feature space (i.e., 32 feature maps back to 64). Finally, the reconstruction layer (v) aggregated the feature maps using 7 by 7 filters to synthesise the output image to the same dimensions as the input.

The first four layers were 2D convolution layers that use the ReLU activation function and a stride of 1. The final reconstruction layer was a 2D deconvolution layer. Zero padding was only adopted in the convolutional layers that have a convolution filter size greater than 1 (i.e., the feature extraction, feature enhancement and reconstruction layers).

Training was performed in Google Colab with a 12 GB RAM. The model was configured with an Adam optimiser and a learning rate of 0.0005. No weight decay was used and the exponential decay rates for the first and second moment estimates were left at their default values of 0.9 and 0.999, respectively.

The ARCNN aims to learn and optimise end-to-end mapping of the input and target images by minimising the difference (i.e., loss) in their intensity values. In this way, the ARCNN learns to suppress block artefacts. We used mean squared error (MSE) or the L2 norm as the loss function for this neural network:

$$Loss = \frac{1}{n} \sum_{i=1}^{n} \| F(Y_i) - X_i \|^2, \tag{5}$$

where $n$ is the batch size, $Y_i$ is the noisy image, $X_i$ is the ground truth/target, $F$ represents the convolutional neural network, and $F(Y_i)$ is the predicted de-noised image.

To train the ARCNN, we obtained clinical MRIs from 35 adult knees (27.7 ± 11.2 years) consisting of a total of 1009 sagittal slices. Ethics approval (reference # AH2627) was granted by the Auckland Health Research Ethics Committee (AHREC). This dataset consisted of three orthogonal MRIs for each subject and was scanned using a proton density fat-suppressed sequence or a SPIR/SPAIR sequence and stored in an NIfTI file format. To generate the training dataset, we applied our SR algorithm to reconstruct each of these sagittal image slices and in the process introduce block artefacts in the image that were characteristic of our algorithm. As a result, we had a dataset of noisy sagittal image slices as well as their corresponding clean versions used as the target for training. We trained the ARCNN through 600 iterations (i.e., epochs) of the entire training dataset with a batch size of 32, and in the end predicted the denoised image to a loss of 0.0010. Further training did not yield significant improvement in loss and so training was terminated after 600 epochs.

The trained ARCNN was then applied to the reconstructed SR image to output a denoised SR image. The final step in the pipeline was to histogram match the denoised SR image to a down-sampled and normalised sagittal image. Histogram matching ensures that the voxel intensity distribution of the denoised SR image closely aligns with that of the original sagittal image, resulting in a more faithful representation and minimises contrast and tone differences that may have been caused by using the ARCNN.

Overall, the novelty of our study primarily lies in the original creation of our SR reconstruction algorithm and the synergistically orchestrated sequence of steps that constitute the entire image enhancement pipeline. The pre-processing and post-processing steps are not entirely novel creations as they are well documented in the literature and have been utilised in isolation before. The pre-processing steps of image registration and re-slicing are routine steps in image processing but serve an important role in aligning the orthogonal imaging stacks. Our linear inverse weighted SR reconstruction algorithm is then employed to amalgamate the information from the orthogonal MR images into a single high resolution image stack. Lastly, the ARCNN derived from Yu et al. [25] is involved in denoising the SR reconstructed images to dampen the artefacts introduced into the image via our SR algorithm. By strategically combining these steps, we are able to improve the resolution of MR images.

## 2.4. Evaluation of Methods

To test the workflow, we obtained a dataset (*n* = 6) of high-resolution (HR) MRIs from 3 subjects using a GE Healthcare 3.0T MR. These images were obtained with an in-plane resolution of 0.31 × 0.31 mm and a slice select resolution of 0.5 mm. A range of sequences including 3D fat-suppressed (FS), water-weighted proton density (PD), PD FS and double echo steady state (DESS) sequences were tested to evaluate the robustness of the workflow (Table 2).

**Table 2.** Dataset used for evaluating the workflow.

| Index | Subject ID | Sequence | MRI Dimensions |
| --- | --- | --- | --- |
| 1 | 1 | Fat | 512 × 512 × 208 |
| 2 | 1 | PD FS | 512 × 512 × 228 |
| 3 | 2 | Fat | 512 × 512 × 208 |
| 4 | 2 | Water PD | 512 × 512 × 208 |
| 5 | 3 | DESS | 512 × 512 × 236 |
| 6 | 3 | Water PD FS | 512 × 512 × 188 |

These image stacks were used to synthesise orthogonal clinical MR imaging stacks of the knee with a high in-plane resolution and low slice select resolution. The synthesised sagittal image had an in-plane resolution of 0.31 × 0.31 mm and a slice select resolution

of 4 mm. The axial and coronal images were synthesised with an in-plane resolution of $0.31 \times 0.5$ mm and a slice select resolution of 5 mm.

We applied the workflow described above to evaluate the output SR reconstructed image and SR denoised image qualitatively and quantitatively with the original high-resolution image. The HR images were down-sampled to in-plane dimensions of $256 \times 256$ and normalised from 0 to 1 to allow direct comparison with the outputs from the workflow.

For quantitative analysis, we computed the mean squared error (MSE), mean error, peak-signal-to-noise ratio (PSNR) and the structural similarity index (SSIM) for the SR denoised images with their original HR image.

$$Mean\ Error = \frac{1}{N} \sum_{i=1}^{n} |H_i - S_i|. \tag{6}$$

The mean error is the average difference between corresponding voxels in the high-resolution and reconstructed images, where $N$ is the total number of voxels in the images, $H_i$ is the voxel intensity at position, i, in the original high-resolution image and $S_i$ is the voxel intensity at position, i, in the SR reconstructed or SR denoised images.

$$MSE = \frac{1}{N} \sum_{i=1}^{n} (H_i - S_i)^2. \tag{7}$$

The mean squared error measures the average squared difference between corresponding voxels in the high-resolution and reconstructed images.

$$PSNR = 10 log_{10} \left( \frac{MAX^2}{MSE} \right). \tag{8}$$

PSNR measures of the voxel-wise differences between images. A higher PSNR value indicates a smaller difference between the high-resolution image and the reconstructed image, implying better image quality. "MAX" is the maximum possible voxel intensity value in the images.

$$SSIM(H,\ S) = \frac{(2\mu_h \mu_s + C_1)(2\sigma_{hs} + C_2)}{(\mu_h^2 + \mu_s^2 + C_1)(\sigma_h^2 + \sigma_s^2 + C_2)}. \tag{9}$$

SSIM compares structural similarity between two images. It takes into account luminance, contrast, and structure. SSIM values range from $-1$ to 1, where 1 indicates perfect similarity. $\mu_h$ and $\mu_s$ are the means of the images, H and S, respectively. $\sigma_h^2$ and $\sigma_s^2$ are the variances of the images, and $\sigma_{hs}$ is the covariance. $C_1$ and $C_2$ are the constants to stabilise the division with a weak denominator [31].

Quantitative analysis was also carried out between the intermediate SR reconstructed and original high-resolution images. Surprisingly, we found that the SR reconstructed image performed better for all quantitative metrics (MSE, mean errors, PSNR and SSIM) compared with SR denoised images. This observation indicates that the ARCNN may induce modifications to the images from their original state. However, it is crucial to note that this discrepancy signifies a trade-off: while there might be a slight compromise in quantitative metrics due to ARCNN's influence, the substantial qualitative improvement in image quality justifies its inclusion in the pipeline. The table of these results is provided in Appendix A.4.

We generated contour plots to evaluate the spatial error distribution over individual image slices. We also generated activation maps to gain further insight into the performance of the ARCNN. These maps visually represent the areas of the input image that exhibit the highest activation by the ARCNN, providing us with a better understanding of the features that the network is focusing on during the denoising process. For this analysis, we used the activation maps from the feature enhancement layer—an intermediate convolution

layer responsible for denoising the extracted features. The corresponding figure displaying the activation maps for a single slice of one of the high-resolution images is presented in Appendix A.5.

To compare the computational expense of reconstructing an SR image, we profiled the code and evaluated the time required to complete each section in the code. This was performed on an Intel i5 processor and an 8 GB RAM. We also determined the time complexity for each section of the SR step code.

## 3. Results

The SR method retained most of the detail from the original sagittal image (Figure 2) while increasing the out-of-plane resolution of the image stack and the number of sagittal slices. Block artefacts were introduced to the SR reconstructed intermediate image as depicted by the voxels that exhibit abrupt intensity changes. These artefacts were particularly noticeable in regions which transitioned from one structure to the next such as from bone to cartilage. The ARCNN from the post-processing step was effective in reducing the blocky artefacts across all sequences tested. This qualitatively improved the texture and detail of the image, almost restoring the original quality of the sagittal image. For example, in the final image, the morphological appearance of the femoral cartilage in both the DESS sequence and Fat MRI sequence appears much more homogeneous with smoother boundaries that were similar to those in the original image when compared with the intermediate SR reconstructed image. This was also supported by what was observed in the activation maps (Appendix A.5). As illustrated in A.5, numerous activation maps showed elevated activation levels along the boundaries of bone, cartilage, and fat areas. These regions coincide with the presence of block artefacts, affirming that the ARCNN is effectively directed towards suppressing these artefacts in the reconstructed image.

After applying the SR method, the axial and coronal views of the sagittal image stacks exhibited improvements in both resolution and qualitative image quality (Figure 3). Similar improvements were observed across images produced by different sequences, although some mild banding artefacts appear to have been introduced into the sagittal stacks in the DESS sequence.

Overall, the mean errors between the final SR denoised image and the original HR image across the six image stacks was $1.40 \pm 2.22\%$, and the minimum error was 0% (Table 3). The output images also had a mean PSNR of 31.94 dB and a mean SSIM of 0.886. The largest MSE, mean errors and standard deviations as well as the lowest PSNRs were present in the fat sequences from Subjects 1 and 2 (Table 3). The greatest errors in intensity value occurred at the boundaries of the bone/cartilage, ligaments, muscles, and fat surrounding the knee, with most of the voxels exhibiting minimal error (Figure 4).

**Table 3.** Reconstruction errors, including the overall MSE, mean error and standard deviation, maximum error, and minimum error between the voxel intensities of the SR denoised image and the original high-resolution image expressed as percentages. The PSNR and SSIM are also provided.

| Subject (MR Sequence) | MSE | Mean Error $\pm$ Standard Deviation | PSNR (dB) | SSIM | Max Error | Min Error |
|---|---|---|---|---|---|---|
| 1 (Fat) | 0.123% | 1.75% $\pm$ 3.04% | 29.11 | 0.897 | 72.4% | 0% |
| 1 (PD FS) | 0.043% | 1.24% $\pm$ 1.68% | 33.62 | 0.880 | 71.4% | 0% |
| 2 (Fat) | 0.130% | 1.89% $\pm$ 3.07% | 28.85 | 0.870 | 70.0% | 0% |
| 2 (Water PD) | 0.037% | 1.12% $\pm$ 1.57% | 34.29 | 0.904 | 62.5% | 0% |
| 3 (DESS) | 0.074% | 1.42% $\pm$ 2.33% | 31.29 | 0.843 | 79.3% | 0% |
| 3 (Water PD FS) | 0.035% | 0.95% $\pm$ 1.63% | 34.50 | 0.919 | 68.5% | 0% |
| Mean | 0.074% | 1.40% $\pm$ 2.22% | 31.94 | 0.886 | 70.7% | 0% |

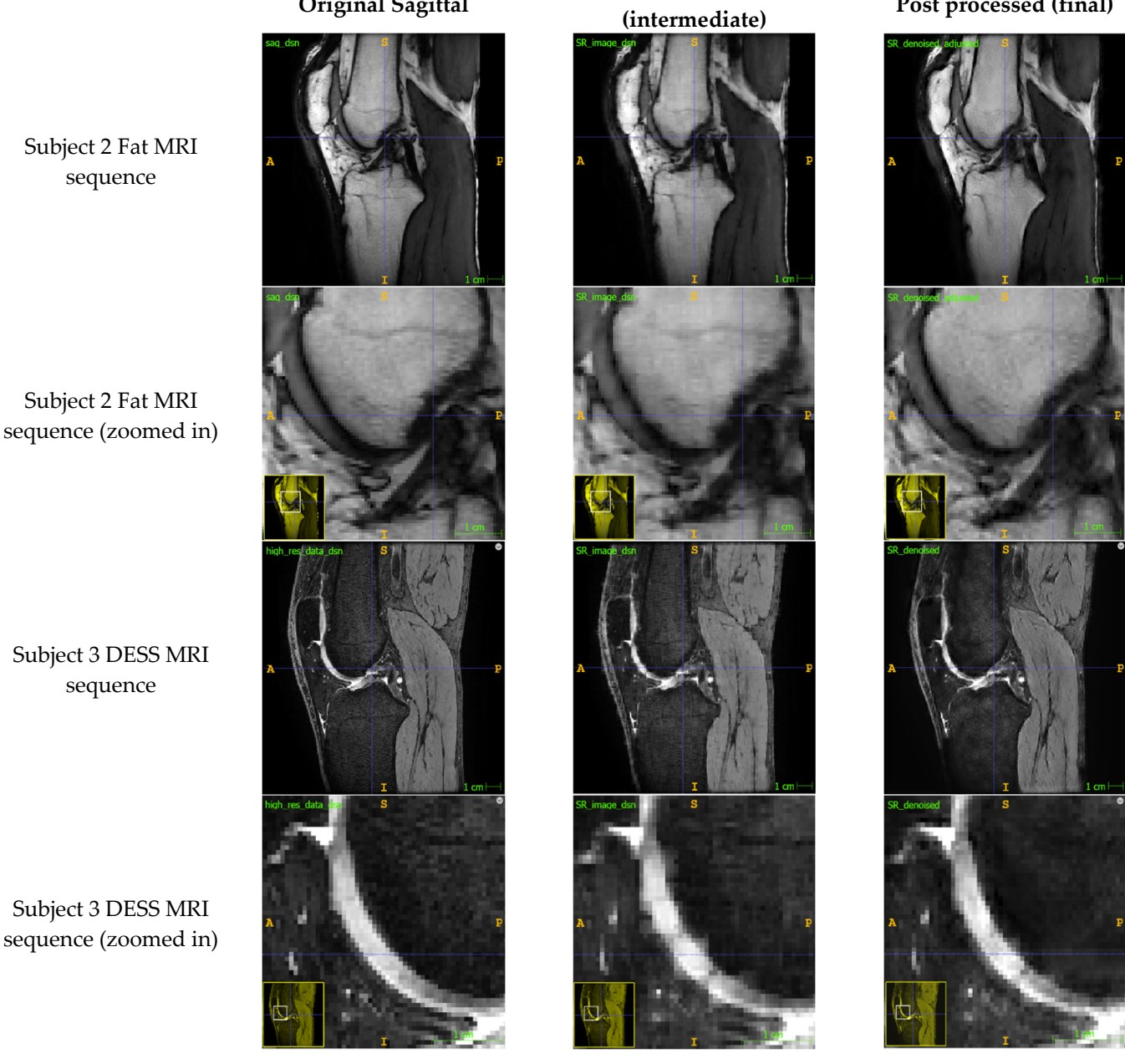

**Figure 2.** Comparison of the original sagittal images to the reconstructed SR images from the SR step (Step 2) and the de-noised images from the ARCNN (Step 3) for two MRI sequences (fat and DESS sequences).

The largest maximum error was 79.3% and occurred in the DESS sequence. The smallest errors were present in the PD sequences with a mean error of $1.10 \pm 0.15\%$, and this was qualitatively reflected in the contour plots (Figure 4).

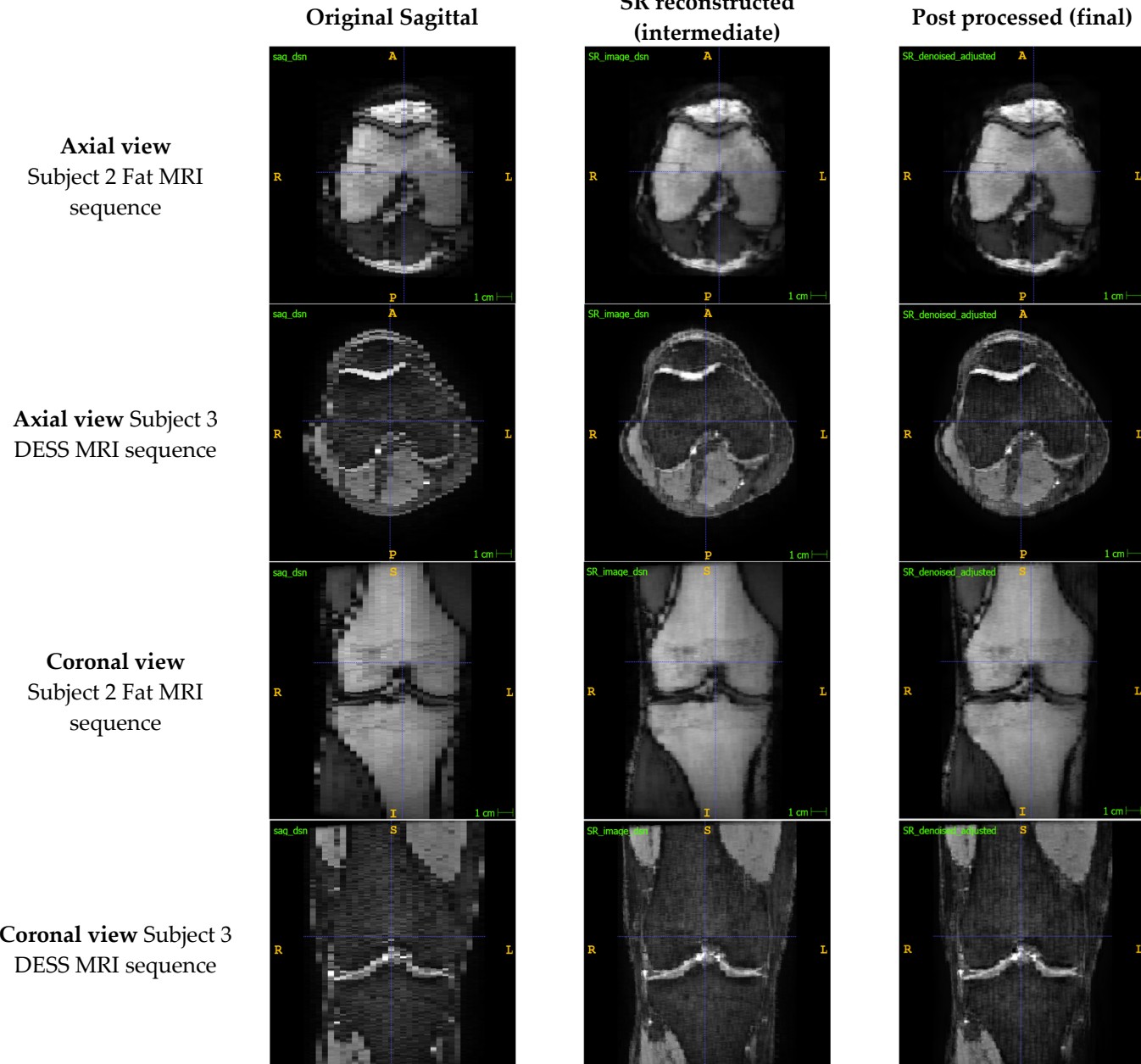

**Figure 3.** Comparison of the out-of-plane axial and coronal views of the sagittal image to the reconstructed SR and de-noised images for two MRI sequences.

The computational cost analysis showed that the SR step takes on average $2725 \pm 569$ s (~45 min) in total to reconstruct the enhanced images (Table 4). The most time-consuming step in the process is creating the binary tree and performing the nearest voxel search which takes 60.9% of the time, followed by initialising the coordinate lists (25.4%) and reconstruction (13.7%). This was supported by the time complexity analysis, where creating the binary tree and then finding the nearest voxels took $O(N\log(N))$ time, whereas initialising the coordinate lists and reconstruction took $O(N)$ time.

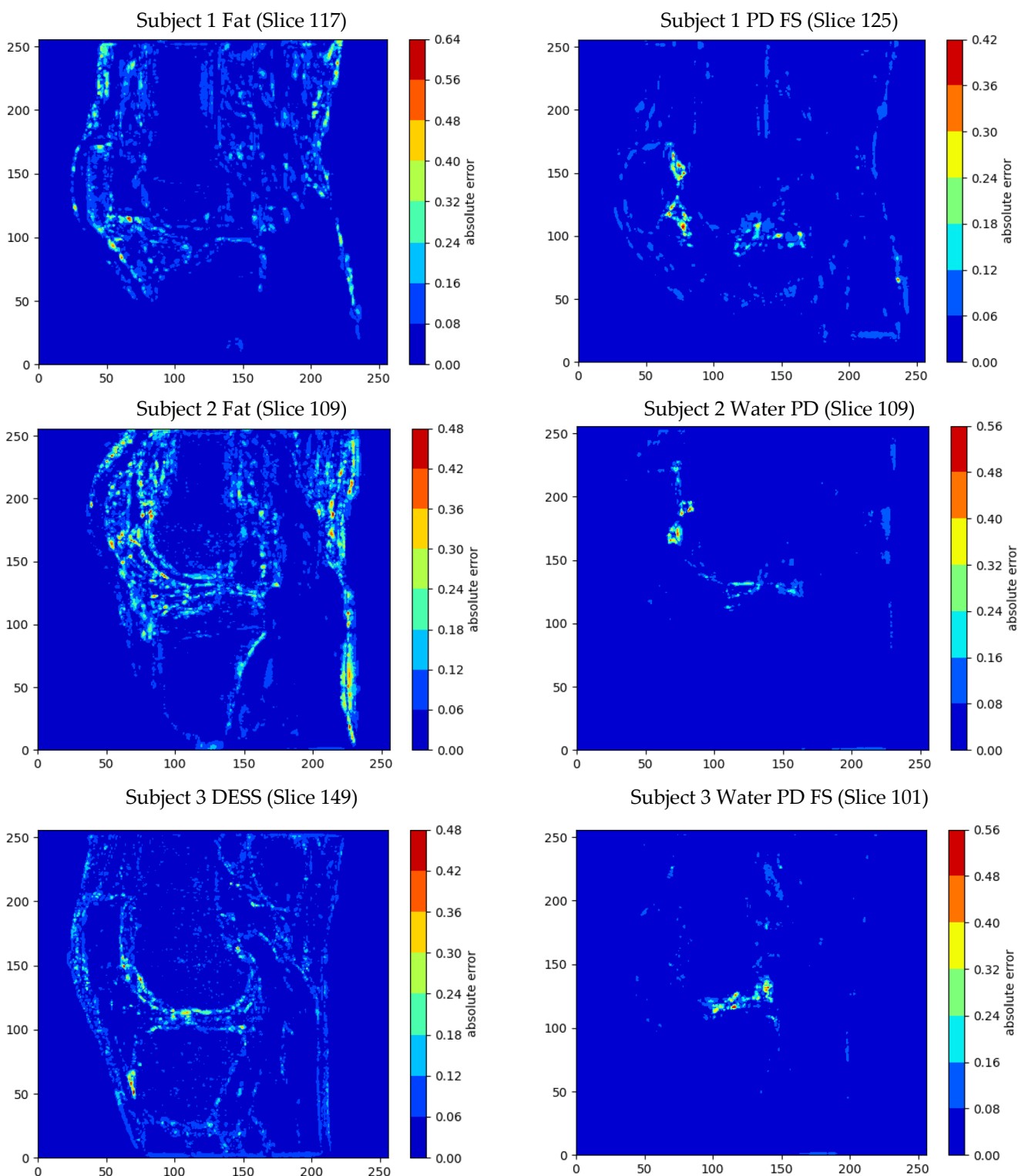

**Figure 4.** Contour plots showing the voxel-wise distribution of errors in the sagittal slices with the largest overall error in intensity value for each of the MRIs.

**Table 4.** Computational cost analysis for the SR step, showing the time taken and time complexity for the 3 parts of this step (initialising coordinate lists, binary tree creation and finding the nearest voxels and reconstruction). NB: N refers to the total number of voxels in the template 3D image array.

| Subject (MR Sequence) | Initialising coordinate Lists (s) | Binary Tree Creation and Finding Nearest Voxels and Their Distances (s) | Reconstruction (s) | Total Time (s) |
|---|---|---|---|---|
| 1 (Fat) | 633 | 1609 | 464 | 2706 |
| 1 (PD FS) | 846 | 2355 | 467 | 3668 |
| 2 (Fat) | 704 | 1519 | 340 | 2563 |
| 2 (Water PD) | 592 | 1487 | 311 | 2390 |
| 3 (DESS) | 842 | 1795 | 372 | 3009 |
| 3 (Water PD FS) | 532 | 1188 | 290 | 2010 |
| Mean | 692 | 1659 | 374 | 2725 |
| Standard deviation | 131 | 394 | 76 | 569 |
| % of total time | 25.4% | 60.9% | 13.7% | 100% |
| Time complexity | $O(N)$ | $O(N\log(N))$ | $O(N)$ | - |

## 4. Discussion

The purpose of this study was to present a hybrid analytical SR pipeline and to quantify the errors in the images produced by the pipeline on a test set of six high-resolution MRIs. We presented an image-domain post-processing SR algorithm that consisted of three steps. The pre-processing step prepared the clinical images for the SR step by re-slicing and co-registering the image stacks. In the SR step, image data from three orthogonal images were combined using a weighting scheme to synthesise a high-resolution image. The post-processing step employed an ARCNN to de-noise any high-frequency block artefacts introduced in the SR step. This pipeline was able to reconstruct a high-resolution isotropic image from three orthogonal images with a high in-plane resolution and low out-of-plane resolution while only introducing a mean error of $1.40 \pm 2.22\%$, showing great promise for clinical utility.

Our SR method improved the out-of-plane resolution while qualitatively retaining most of the detail from the original sagittal images (Figures 2 and 3). In our test set of six high-resolution MRIs, we found a mean error of $1.40 \pm 2.22\%$ across the different sequences, and a mean of $1.10 \pm 0.15\%$ across the PD images, which reassured that the method was closely preserving the details in the images. This was further supported by the output images having a mean PSNR and SSIM of 31.94 dB and 0.886, respectively, demonstrating that the method was significantly minimizing information loss and producing high-fidelity images. In comparing our method to the existing literature, it is imperative to note the inherent challenges in finding directly comparable studies. Variations in voxel thickness, resolution, MR sequences, and anatomical structures across different datasets hinder a straightforward comparison. However, we can still acquire some valuable insights. Compared to other SR methods in the literature [5–24], we did not use interpolation, statistical regression, or machine learning to generate voxel intensity values. Instead, voxel intensities were calculated analytically with a linear inverse weighting scheme and the ARCNN was only used to de-noise the high-frequency errors. Avoiding interpolation and restricting machine learning to only the post-processing step meant that our output images were devoid of blurring and overfitting errors that other methods sometimes experience [5,18–24].

In Figure 3, we can see that the SR step improved the out-of-plane resolution of the sagittal MRI with only a slight decrease in the in-plane sagittal image quality. Through SR, we took a sagittal image with 25–40 slices and reconstructed an image that had over 200 slices. After denoising the image with the ARCNN, the resolution was maintained, though we did observe mild bandings in the image for the DESS MRI. This is because

the ARCNN only denoised the image in the sagittal orientation without any regard for how it impacts out-of-plane image quality. This is an area we can look to improve in by developing a 3D ARCNN that volumetrically denoises the SR reconstructed image in 3D and thus further enhances output image quality.

Our SR method appeared to be robust to different MRI sequences (Table 3 and Figure 4) and we saw improvements in all the images. This distinguishes our method from others in the literature [7,15–17,23] which were only tested on a single MRI sequence, and so it is undetermined how well their methods perform with other MRI sequences. However, the PD MR sequences showed lower errors in intensity value and higher PSNR than the non-PD sequences. This was likely due to the training of the ARCNN, which was only trained on SPIR/SPAIR and PD FS sequences and not on fat MR sequences. As such, the ARCNN did not perform as well when denoising images that had considerably different appearances than it was trained to expect. Expanding the training dataset with data from additional subjects and other types of MR acquisition sequences would significantly help to reduce the errors.

The contour plots (Figure 4) highlighted that the majority of the SR image exhibited a low error, with only a few regions of voxels exhibiting an error. Most of the error in intensity value occurred at the boundaries of structures (e.g., bone, cartilage, muscles, and fat). There are a couple of factors that may explain these observations. First, these were the areas where we saw the most block artefacts in the SR reconstructed image (Figure 2). These artefacts are most prevalent at areas of high-intensity change, such as from bone to cartilage, or bone to fluid, and may arise due to partial volume effects. It is our understanding that in calculating the new voxel intensity values, these high intensity changes were preserved. As such, we hypothesise that the acquisition of more data from additional orientations may help to reduce these artefacts. Second, these regions exhibited larger errors because this was where the training and test sequences differed the most in contrast. We expect that further training of the ARCNN, by including more images and MRI sequences, will reduce these errors further.

The results of this paper should be considered in the context of the following limitations. Computational efficiency is a particular limitation of the pipeline. More specifically, the SR step is by far the most time-consuming step in the workflow, taking on average 45 min to reconstruct an SR image with around 200 slices (Table 4). In total, 61% of this time is used to create the binary tree and find the nearest voxels and their corresponding distances, whereas 25% and 14% of the time are used for initialising the coordinate lists and reconstruction, respectively. Image domain methods, like the one we propose, tend to have more of a computational burden as they can often take around 3 h to reconstruct the image [1]. Frequency domain methods can also take around 1 h to reconstruct [1]. Machine learning methods can vary significantly in computation time from 10 s to up to 3 h [20], whereas interpolation methods tend to be consistently very efficient [21,23,24]. Our time of around 45 min is significantly quicker than the times of other image domain methods, and although it may be slower than some other methods, it is serviceable for clinical and research use. Computer RAM also constrains this step as the KD tree Python function used to construct the binary tree and find the closest voxels and their distances, requiring the coordinates of all the images to be inputted as lists. Having coordinate lists for the histogram matched axial, coronal, original sagittal, and the template sagittal image consumes a lot of RAM, which can limit the possible size of the SR image. Training the ARCNN is another computationally expensive step as the large MRI sizes make training a slow process. However, once the network is trained, the model is quick to denoise the reconstructed images. One thing to note is that the computational results presented here are likely overstated by the hardware we used when testing the workflow. In a clinical setting, there is likely to be more computational power and memory capacity, which will make the process more efficient. Further computational efficiency could also be achieved by using a lower-level programming language such as C, something that could be implemented in future work.

While our study demonstrated promising results (Table 3, Figures 2–4), it is essential to acknowledge the inherent limitation associated with the relatively small testing dataset, consisting of only six knee MRIs. The constrained sample size raises valid concerns about the generalizability and robustness of our proposed method across a broader range of imaging scenarios such as varying voxel thicknesses and intensities. It is crucial to mention that the selection of this dataset was dictated by the availability of advanced, high-resolution MR images necessary for our unique methodology. Nevertheless, the confined dataset does limit the scope of our conclusions. To ensure a more comprehensive assessment of our method's performance, future investigations will look to increase the testing sample size while incorporating a more diverse set of imaging data which encompasses a variety of knee morphologies and imaging sequences. The insights gained from a more extensive and varied dataset will not only bolster the reliability of our findings, but also contribute to a deeper understanding of the method's performance characteristics and utility.

The SR method overcomes the limitations of MRI and thus has tremendous potential for clinical and research purposes, reducing the need to purchase new and expensive equipment. The improved out-of-plane resolution of the output images allows for better visualisation than the original image; clinicians can also gain access to enhanced images for diagnostic purposes or for surgical planning. The output images allow better visualisation of small structures such as ligaments like the ACL, enabling researchers to generate more anatomically accurate models of the human body for scientific exploration. Although this research focused on the knee, the workflow can be applied to images of other anatomical structures and so also has potential applicability outside orthopaedics.

## 5. Conclusions

In this study, we introduced a hybrid analytical super-resolution (SR) pipeline to enhance MRI scans without costly hardware upgrades. Our approach effectively improved out-of-plane resolution while preserving image quality. Validation on high-resolution knee MRIs showcased promising results with a mean error of $1.40 \pm 2.22\%$, a mean PSNR of 31.94 dB and a mean SSIM of 0.886 between the SR denoised and original images.

The pipeline displayed robustness across various MRI sequences and avoided issues like blurring or overfitting associated with interpolation or extensive machine learning. Despite being computationally intensive (~45 min per SR reconstruction), it offers a cost-effective alternative to hardware upgrades.

Future work could focus on optimizing computational efficiency without compromising accuracy, potentially exploring lower-level programming languages or parallel processing. Moreover, refining the post-processing artefact reduction step through developing a 3D ARCNN for volumetric denoising could further enhance image quality. Expanding the testing sample size with the inclusion of a diverse range of MR images is also important to strengthen the reliability of our results and provide a deeper understanding of the performance of our proposed pipeline.

This method holds significant potential for clinical use, aiding in accurate diagnosis and surgical planning while also enabling more precise anatomical modelling for research purposes. Overall, our hybrid SR pipeline presents a promising avenue for enhancing MRI resolution without substantial hardware investments, contributing to advancements in medical imaging and clinical applications.

**Author Contributions:** Conceptualization, V.P., A.W. and M.T.-Y.S.; methodology, V.P., A.W. and M.T.-Y.S.; software, V.P. and M.T.-Y.S.; validation, V.P. and M.T.-Y.S.; formal analysis, V.P.; investigation, V.P., A.W. and M.T.-Y.S.; resources, A.W., M.T.-Y.S. and A.P.M.; data curation, V.P. and M.T.-Y.S.; writing—original draft preparation, V.P. and M.T.-Y.S.; writing—review and editing, V.P. and M.T.-Y.S.; visualization, V.P. and M.T.-Y.S.; supervision, A.W., M.T.-Y.S. and A.P.M.; project administration, A.W., M.T.-Y.S. and A.P.M.; funding acquisition, A.W. and M.T.-Y.S. All authors have read and agreed to the published version of the manuscript.

**Funding:** This research was supported by the Auckland Bioengineering Institute Summer Research Studentship.

**Institutional Review Board Statement:** Ethics approval (reference # AH2627) was granted by the Auckland Health Research Ethics Committee (AHREC) for the MR image dataset used for training the ARCNN.

**Informed Consent Statement:** Informed consent was obtained from all subjects involved in the study.

**Data Availability Statement:** The MRI datasets utilised and presented in this article are not readily available in order to safeguard patient confidentiality and uphold ethical standards in medical research.

**Conflicts of Interest:** The authors declare no conflict of interest.

## Appendix A.

*Appendix A.1. Presenting the Comparison between SR Images Reconstructed with Two Planes vs. Three Planes*

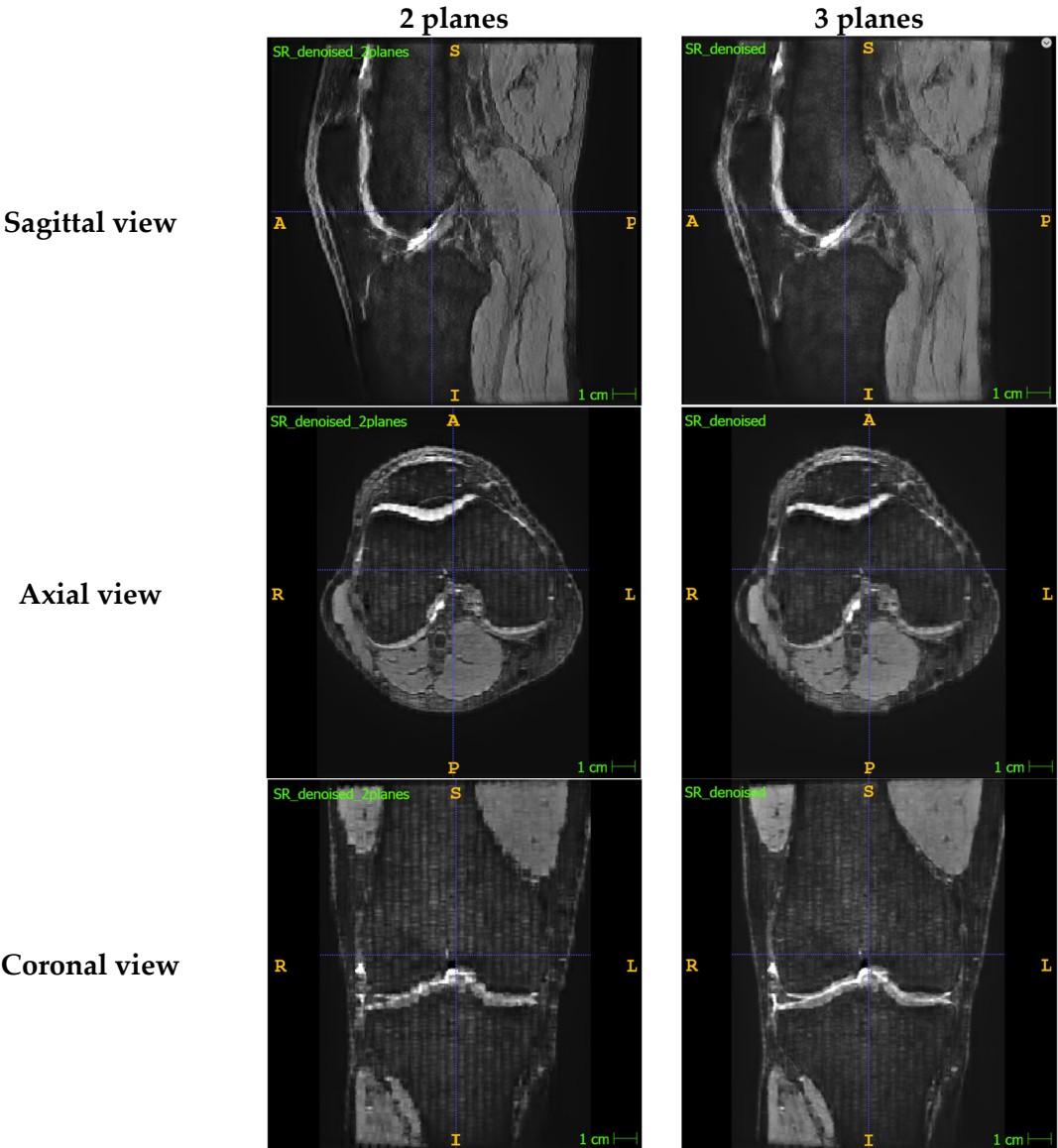

**Figure A1.** Comparison of SR denoised images that were reconstructed with two MR image planes (i.e., using the sagittal and axial images) vs. with three MR image planes (i.e., using the sagittal, axial and coronal images). All three views of the SR denoised images are provided.

As shown in the above figure, the denoised SR image reconstructed from two planes has lower image quality than the denoised SR image reconstructed from all three orthogonal planes. This is noticeable in all three views but especially in the view corresponding to the plane that was not included during reconstruction (i.e., coronal plane), which significantly suffers with more banding and artefacts around the boundaries of the structures.

*Appendix A.2. Summary of the 3D Slicer Registration Parameters*

**Table A1.** Table presenting the 3D slicer registration parameters.

| Input Images | |
|---|---|
| Fixed image volume | Isotropic sagittal image |
| Moving image volume | Axial image OR coronal image |
| Percentage of samples | 0.002 |
| B-spline grid size | 14,10,12 |
| Output settings | |
| Slicer linear transform | None |
| Slicer Bspline transform | None |
| Output image volume | Registered axial image OR Registered coronal image |
| **Transform initialisation settings** | |
| Initialisation transform | None |
| Initialise transform mode | Off |
| Registration phases | Rigid and affine selected |
| **Image Mask and Pre-processing** | Default settings |
| **Advanced output settings** | |
| Fixed image volume 2 | None |
| Moving image volume 2 | None |
| Output image pixel type | Float |
| Background fill value | 0.0 |
| Interpolation mode | Linear |
| **Advanced optimisation settings** | |
| Max iterations | 1500 |
| Maximum step length | 0.05 |
| Minimum step length | 0.001 |
| Relaxation factor | 0.5 |
| Transform scale | 1000.0 |
| Reproportion scale | 1.0 |
| Skew scale | 1.0 |
| Maximum B-spline displacement | 0.0 |
| **Expert-only parameters** | |
| Fixed image time index | 0 |
| Moving image time index | 0 |
| Histogram bin count | 50 |
| Histogram match point count | 10 |

**Table A1.** *Cont.*

| Input Images | |
|---|---|
| Cost metric | NC (i.e., normalised correlation) |
| Inferior cut off from centre | 1000.0 |
| ROIAuto dilate size | 0.0 |
| ROIAuto closing size | 9.0 |
| Number of samples | 0 |
| Stripped output transform | None |
| Output transform | None |
| **Debugging parameters** | Default settings |

*Appendix A.3. Worked Example Showing How the Intensity Value Is Calculated for a Template Voxel*

**Table A2.** Table showing the distances from the template voxel and intensity values for each of the three hypothetical MRI image stacks.

| Image Stack | Distance from Template Voxel (mm) | Intensity Value |
|---|---|---|
| Sagittal | 2 | 173 |
| Axial | 3 | 198 |
| Coronal | 5 | 127 |

$$\text{Using (4)}, \ d_{\text{Total}_{ijk}} = 2 + 3 + 5 = 10,$$

$$\text{Using (2)}, \ w_{\text{sagittal}} = \frac{10 - 2}{20} = 0.4$$

$$\text{Using (2)}, \ w_{\text{axial}} = \frac{10 - 3}{20} = 0.35,$$

$$\text{Using (2)}, \ w_{\text{coronal}} = \frac{10 - 5}{20} = 0.25.$$

$$\text{Using (1)}, \ v_{\text{template}_{ijk}} = (0.4 \times 173) + (0.35 \times 198) + (0.25 \times 127) = 170.25.$$

*Appendix A.4. Presenting the Quantitative Analysis Results between the Intermediate SR Reconstructed Image and the Original High-Resolution MR Image*

**Table A3.** Reconstruction errors, including the overall MSE, mean error and standard deviation, maximum error, and minimum error between the voxel intensities of the SR reconstructed image and the original high-resolution image expressed as percentages. The PSNR and SSIM are also provided.

| Subject (MR Sequence) | MSE | Mean Error $\pm$ Standard Deviation | PSNR (dB) | SSIM | Max Error | Min Error |
|---|---|---|---|---|---|---|
| 1 (Fat) | 0.091% | 1.36% $\pm$ 2.70% | 30.40 | 0.939 | 58.8% | 0% |
| 1 (PD FS) | 0.013% | 0.50% $\pm$ 1.03% | 38.80 | 0.966 | 62.6% | 0% |
| 2 (Fat) | 0.103% | 1.47% $\pm$ 2.86% | 29.86 | 0.926 | 64.8% | 0% |
| 2 (Water PD) | 0.019% | 0.76% $\pm$ 1.15% | 37.20 | 0.965 | 61.3% | 0% |
| 3 (DESS) | 0.058% | 1.14% $\pm$ 2.12% | 32.36 | 0.888 | 77.3% | 0% |
| 3 (Water PD FS) | 0.027% | 0.85% $\pm$ 1.39% | 35.76 | 0.966 | 60.4% | 0% |
| Mean | 0.052% | 1.01% $\pm$ 1.88% | 34.06 | 0.942 | 64.2% | 0% |

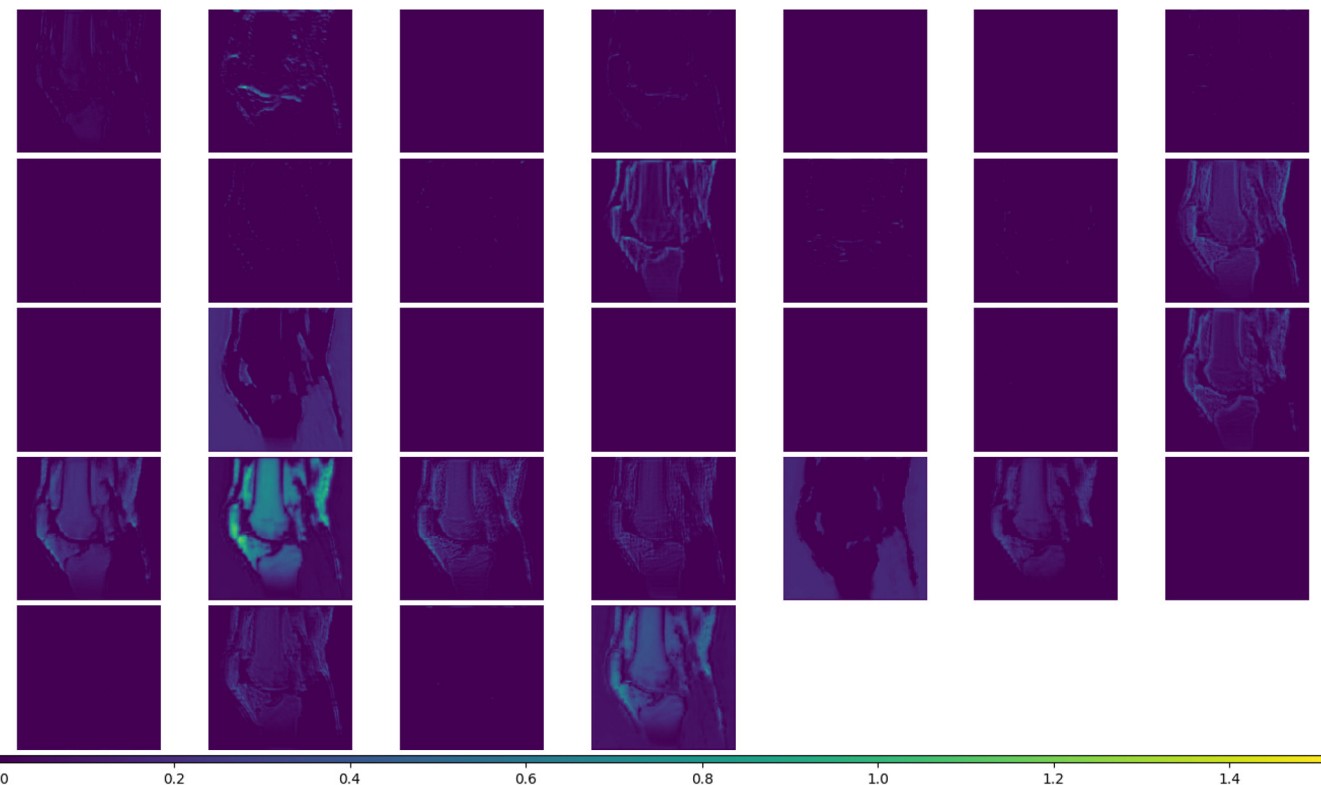

**Figure A2.** Figure Showing All 32 Activation Maps for the "Feature Enhancement" Layer of the ARCNN for Subject 1 Fat MRI, Image Slice #117.

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
