# Peer review of "Enhancing Knee MR Image Clarity through Image Domain Super-Resolution Reconstruction"

_bioengineering, doi:10.3390/bioengineering11020186_

Round 1
Reviewer 1 Report (Previous Reviewer 1)
Comments and Suggestions for Authors
The paper has been effectively presented.
The CNN architecture is well-crafted; however, could you elaborate on the uniqueness of your work and provide a detailed comparison with existing literature?
While the paper is quite commendable, I suggest including Class Activation Maps (CAM) figures for the CNN to illustrate how the proposed architecture improves image resolution.
Additionally, could you share the dataset for the sake of reproducibility and further research productivity?
Author Response
Please see the attachment

Reviewer 2 Report (Previous Reviewer 3)
Comments and Suggestions for Authors
Prior comment: 'The results are based only with 6 images and so they were not robust or generalizable across various thickness or intensity wise. Also Author could apply his model to various dataset publicly available and produce the images to show the output.' is important to know the potential of method. Author may add further data from own study or public domain data to validate his results. With limited data the proposed method potential can't be verified.
Author Response
Please see the attachment

Reviewer 3 Report (Previous Reviewer 4)
Comments and Suggestions for Authors
This is a revised manuscript based on a previously reviewed paper. All my concerns have been addressed.
Figure 4: It would be good to use the same color scaling (vertical bar) to visualize these images.
A reference number could be added for the equation (9).
NA
Author Response
Please see the attachment

This manuscript is a resubmission of an earlier submission. The following is a list of the peer review reports and author responses from that submission.
Round 1
Reviewer 1 Report
Comments and Suggestions for Authors
The authors made a significant contribution to biomedical imaging, but several points need clarification and elaboration:
- A comprehensive description of the database used, including details about its source.
- Clearer explanation of the training options and epochs for the proposed ARCNN.
- Details about the experimental setup, such as the software programs used and specifications of the computer station.
- Explanation regarding the acquisition of clean images for comparison as the target for the designed ARCNN.
- Clarification about the purpose behind alternating between filter sizes of 1*1 and 7*7.
- Suggestion to include a network diagram in the image for better elucidation.
- Highlighting the primary novelty of their work.
- Request to compare the obtained results with existing studies for benchmarking purposes.
- Suggestion to include a conclusive section summarizing the findings.
- Request for sharing the code and data, especially pertaining to the ARCNN, including clean and noisy images for training and testing the network.
- Exploration of the possibility to adapt the proposed network to work with 3D images instead of processing individual slices.
- What are the limitations of your work.
Can you increase your sample size
Reviewer 2 Report
Comments and Suggestions for Authors
1) The drawbacks of the existing methods must be illustrated in the introduction section.
2) What is the rationale behind choosing the loss function as indicated in eqn (5)? Will another loss function changes the performance measures?
3) The methodology section seems to be highly theorotical. Few mathematical modelling should be included.
4) What is the reason for the difference in results for different cases/subjects?
5) Is error alone enough to justify the level of increase in the quality of the input images?
6) How did you estimate the time? Will it be different for a computer with different specification? Have you done any computational complexity analysis?
7) A comparative analysis with the existing methods must be given.
Comments on the Quality of English Language
moderate changes required.
Reviewer 3 Report
Comments and Suggestions for Authors
This study developed a hybrid analytical super-resolution (SR) pipeline to enhance the resolution of medical MRI scans, aiming to bypass the need for expensive hardware upgrades. The pipeline, validated on six high-resolution knee MRIs, involves pre-processing, SR reconstruction combining three orthogonal image stacks, and post-processing with an artefact reduction convolutional neural network. It showed a low mean error and improved resolution, proving effective across various MRI sequences and suggesting potential for broader clinical use in improving diagnostics and anatomical modeling. I have few concerns and they were mentioned below.
1.The results are based only with 6 images and so the they were not robust or generalizable across various thickness or intensity wise. Also Author could apply his model to various dataset publicly available and produce the images to show the output.
2. Author should compare the results with standard methods like SSIM and PSNR.
3. There are many publications available with superresolution and author should compare and contrast his model output against others (https://doi.org/10.1038/s41598-022-10298-6; DOI: 10.3390/tomography8020073)
Reviewer 4 Report
Comments and Suggestions for Authors
This paper proposes a hybrid SR approach. It consists of three modules, including pre-processing to re-slice and register the image stacks; SR reconstruction to combine information from three orthogonal image stacks to generate a high-resolution image stack; and post-processing using an artifact reduction convolutional neural network (ARCNN).
Since the proposed approach is a hybrid approach, an ablation study is required to verify the contribution from each module. For example, in Figures 2 and 3, a quantitative performance comparison is needed to compare the image result with post-processing and that without post-processing.
In addition, some details of the proposed approach need to be further clarified.
1. Equation (1): How is the summation performed? The indices i, j, and k need to be described.
2. Table 1: Is this network architecture the same as that proposed in the reference [25]? If so, a reference number needs to be added for Table 1. Otherwise, the modification needs to be highlighted.
3. Is there any augmentation performed during the training?
NA
